Measuring the well-being of health care professionals in the Punjab: a psychometric evaluation of the Warwick–Edinburgh Mental Well-being Scale in a Pakistani population

Waqas Ahmed 1 ahmedwaqas1990@hotmail.com
Ahmad Waqas 1
Haddad Mark 2
Taggart Frances M. 3
Muhammad Zerwah 1
Bukhari Muhammad Hamza 1
Sami Shahzad Ahmed 1
Batool Sayyeda Mehak 1
Najeeb Fiza 1
Hanif Ayesha 1
Rizvi Zehra Ali 1
Ejaz Sumbul 1
1 CMH Lahore Medical College and Institute of Dentistry , Lahore , Pakistan
2 Centre for Mental Health Research, School of Health Sciences, City University London , London , United Kingdom
3 Division of Health Sciences, Medical School, University of Warwick , Coventry , United Kingdom
Coyne James
Electronic publication date: 2015 Oct 1
Publication date: 2015
Volume: 3
Electronic Location ID: e1264
Received 2015 May 15; Accepted 2015 Sep 3
Copyright: © 2015 Waqas et al.
Copyright year: 2015
Copyright holder: Waqas et al.
License: This is an open access article distributed under the terms of the Creative Commons Attribution License, which permits unrestricted use, distribution, reproduction and adaptation in any medium and for any purpose provided that it is properly attributed. For attribution, the original author(s), title, publication source (PeerJ) and either DOI or URL of the article must be cited.
License URL: https://creativecommons.org/licenses/by/4.0/

Keywords: Burnout, WEMWBS, Warwick–Edinburgh Mental Well-being Scale, Mental fatigue, Health personnel, Mental health

Funding: The authors declare that this study was not funded by any agency.

==============================
Background. There is growing awareness of the public health importance of mental well-being both in the general population and in specific groups. The well-being of health professionals is likely to influence the quality of the care they deliver. This study was carried out to examine the well-being of Pakistani healthcare professionals, and to evaluate the psychometric performance of the Warwick–Edinburgh Mental Well-being Scale (WEMWBS) in this population.

Methods. A cross-sectional survey was carried out from June, 2013 to December, 2014 among 1,271 Pakistani health care providers (HCPs) working in seven different cities in Punjab province, Pakistan, to examine the acceptability, internal consistency, test-retest reliability and content and construct validity of the English version of the WEMWBS in a Pakistani population sample. All data were analyzed in SPSS v. 21.

Results. Our analysis demonstrated unidimensional construct validity, high internal consistency (0.89) and test-retest reliability, good validity and easy readability of WEMWBS in our sample of Pakistani HCPs. The mean WEMWBS score was 48.1 (SD 9.4), which is lower than in the general population in other countries. Male HCPs scored significantly higher on the WEMWBS than their female counterparts (P < 0.05), and older respondents had higher scores.

Conclusion. The WEMWBS appears acceptable for use in Pakistani HCPs, and findings from this study verify its validity and internal consistency for this population sample. Our respondents had lower well-being scores than those reported in general population surveys in the UK.

Introduction

Subjective or psychological well-being is increasingly recognized as a crucial aspect of overall health that highlights the importance of positive mental health. It relates to the classic World Health Organization (WHO) definition of health as more than just the absence of disease: “Health is a state of complete physical, mental and social well-being and not merely the absence of disease or infirmity” (World Health Organization, 1946). The importance of mental health for overall well-being is emphasized in international policy as the “foundation for well-being and effective functioning for both the individual and the community (which) allows individuals to realise their abilities, cope with the normal stresses of life, work productively and fruitfully, and make a contribution to their community” (World Health Organisation, 2004; OECD, 2013).

Subjective well-being has a long tradition as a key part of the conceptualization of quality of life (OECD, 2013) and its measurement has been embedded within this construct. For example, measures of quality of life and of health-related quality of life commonly incorporate questions concerning positive and negative affect (World Health Organization , 1998; McHorney et al., 1994) as well as life satisfaction and considerations of fulfilment and purpose, as for instance in the triennial European Quality of Life Survey (Eurofound, 2012a). However, over the past two decades, interest in measuring subjective well-being has expanded, with the emergence of evidence that it can be measured in ways that are valid and reliable (OECD, 2013). There is now widespread acknowledgement that focusing on the measurement of subjective well-being is an essential part of measuring quality of life that merits specific measures and approaches (Stiglitz, Sen & Fitoussi, 2009).

This focus on the positive elements of mental health and functioning is superseding a previous emphasis on deficits, problems and symptoms. This is evident within psychiatric research, mental health policy and clinical practice (Slade, 2010), where the concepts of recovery and the use of well-being measures are increasing (Warner, 2009). Within the broader context of public health and policy, the current focus on positive elements relates to an acknowledgment that promoting well-being and independence is essential to prevent ill health, which is of particular relevance in light of the growing burden of long-term conditions (Nolte & McKee, 2008).

Within health services, staff well-being is recognized to be relevant to the quality and safety of health care delivery (Royal College of Physicians, 2015) as well as to efforts and expenditures to recruit and retain clinical staff (Boorman, 2009). The health care sector ranks high in studies of important potential causes of work stress (Eurofound, 2012b), with health professionals (in particular nurses) among those who report the highest 3-year average rates of work-related stress, depression or anxiety (Health and safety executive, 2015). An extensive literature has explored stress among health professionals, using measures of psychological distress as well as more conceptually elaborate and specific tools such as burnout instruments (McManus et al., 2007). Stressors that may be evident in health professionals’ work environment include long and unsocial working hours, prolonged and cumulative involvement with seriously ill patients and their relatives, and exposure to organizational conflicts. Even during their training, health care providers (HCPs) are exposed to a plethora of academic, psychosocial and health-related stressors (Waqas et al., 2015). Medical students and junior doctors have been reported to be at particular risk, but estimates of prevalence vary widely from 18% to 82% (Prins et al., 2007).

Studies among health care professionals in Pakistan have reported a high prevalence of anxiety, depression, high stress levels and burnout, highlighting the need to address the issues of poor job satisfaction, sleep deprivation due to shift patterns, patient overload, authoritarian work relationships and limited peer and social support (Aslam, Mansoor & Suleman, 2013; Usman et al., 2013; Fradelos et al., 2014). In line with the growing emphasis on measures of positive health rather than a focus on deficits and problems, we believed it would be useful to measure the subjective mental well-being of HPCs in Pakistan. The most widely known measure of mental well-being in the UK is the Warwick Edinburgh Mental Well-being Scale (WEMWBS), developed by the Universities of Warwick and Edinburgh and funded by NHS Scotland.

The current study was devised to examine the psychometric characteristics of the WEMWBS instrument among English-speaking Pakistani health professionals. This information is useful because (i) few psychological and psychiatric rating scales have been cross-culturally validated for use in the Pakistani population (Ahmer, Faruqui & Aijaz, 2007), and (ii) the mental health scales that have been evaluated for use in this population are largely measures of symptoms of mental disorders rather than of positive mental health.

Materials and Methods

Questionnaire and instrument

The questionnaire consisted of two sections. The first section recorded information related to respondents’ demographics and profession, and the second section assessed mental well-being levels with the WEMWBS.

The WEMWBS was developed and extensively validated in the UK general population (Tennant et al., 2007). This self-administered questionnaire comprises 14 positively-worded items that assess eudaimonic (focused on psychological functioning and a sense of meaning and purpose in life) and hedonic (focused on happiness, contentment and life satisfaction) constructs of mental well-being. Responses are recorded on a 5-point Likert scale ranging from “none of the time” (score 1) to “all the time” (score 5). For analysis, the total score is obtained by summing the scores for all of the items. There is no reverse coding. Higher scores indicate greater well-being. The lowest possible score is 14 (low well-being) and the highest possible score is 70 (high well-being).

Use in the UK and in other countries (Castellví et al., 2014) showed that the scale has good psychometric properties, and is feasible, reliable, and sensitive to change. Cronbach’s alpha for the scale lies between 0.87 and 0.91, and 1-week test-retest reliability studies reported very good correlation with r = 0.83. Validation studies have been done in different population samples in the UK, such as Scottish undergraduate and postgraduate students, the general population (Tennant et al., 2007), English and Scottish teenagers (Clarke et al., 2011), and Pakistani and Chinese ethnic minorities (Taggart et al., 2013). The WEMWBS has also demonstrated high internal consistency, good content and face validity, normal response distribution and no floor or ceiling effects in previous validation studies (Tennant et al., 2007; Clarke et al., 2011; Taggart et al., 2013)

Pilot survey

A pilot study was conducted in which a convenience sample of 50 health care professionals including doctors, nurses and dentists in the Combined Military Hospitals in Lahore completed and commented on the WEMWBS to tell us whether the original English-language version could be easily understood and completed. Positive feedback from these participants indicated that the WEMWBS was very easy to understand. We therefore felt it was not necessary to translate this instrument into Urdu (the participants’ first language).

Sample size calculation

Sample size is generally calculated on the basis of variability in the sample and the expected effect size. Usually both of these are unknown before starting the study, and numbers are estimated from previous studies, or sometimes with “rules of thumb” based on previous experience. Most studies of questionnaires validation in social sciences use 5 to 10 respondents per questionnaire item for factor analysis. Factor analyses are subject to the strength of communalities and factor loading, so these statistical tests should instead be based on large samples for reliable results (MacCallum et al., 1999). Comrey and Lee recommend at least 500 cases, and according to their rating scale a sample size of 1,000 or more is considered excellent for factor analysis studies (Lee, 1992). We therefore aimed for a sample size of at least 1,000 HCPs.

Procedure

This cross-sectional survey was carried out from June 2013 to December 2014. The study sample consisted of doctors, nurses, physiotherapists, pharmacists and dentists practicing either in hospital-based or private settings in various cities of the province of Punjab, Pakistan. Seven districts—Lahore, Faisalabad, Gujrat, Multan, Rawalpindi, Islamabad and Sheikhupura—were included in the survey for a study sample representative of the Punjabi population. However, due to resource limitations, we could not obtain a systematic random sample. Data were therefore collected with a convenience sampling approach. Sixteen medical students (currently in their final year of the MBBS degree program) approached HCPs in all available clinical settings in the Punjab province. An independent sample of 50 HCPs from CMH Lahore participated in the test-retest assessment of WEMWBS and completed the questionnaire twice, with a test-retest interval of 7 days.

Ethical approval was obtained from the Ethical Review Committee of CMH Lahore Medical College, Lahore Cantt, Pakistan. Written informed consent was provided by all the study participants. The respondents were ensured anonymity and informed that only group-level findings would be reported.

Statistical analysis

All data were analyzed with SPSS v.21 (IBM Chicago, IL, USA). Frequencies and descriptive statistics were calculated for demographic variables and total scores on the WEMWBS. Histograms and quantile–quantile (Q–Q) plots were visualized to check the assumption of normality for WEMWBS scores, and we also looked for floor and ceiling effects in the response distribution. Factor analysis with principal component analysis (PCA) and quartimax rotation were done to evaluate the construct validity of the WEMWBS. Prior to PCA, its suitability was assessed with the following criteria: correlation coefficient >0.3 for all variables, an overall Kaiser–Meyer–Olkin (KMO) measure greater than 0.6, and a statistically significant Barlett’s test of sphericity (P < 0.05). The number of components to retain was determined with Cartell’s scree plot, eigenvalues >1, interpretability criterion, the amount of variance explained and reliability analysis.

Only those statements were included that had a factor loading >0.3. Cronbach’s alpha coefficient was calculated to estimate the internal consistency of WEMWBS in our study sample, and a value of 0.70–0.9 was considered acceptable (Streiner, 2008). Item-total correlations were calculated with Pearson’s product moment correlation coefficient, and values that were substantial but not excessive (greater than or equal to 0.2 and less than 0.8) were sought (Streiner, 2008). Test-retest reliability was evaluated as the intraclass correlation coefficient with a two-factor mixed effects model and type consistency.

Floor and ceiling effects were sought by examining response patterns for each of the factors derived from factor analysis. Scores were graphed as a histogram, and the distribution of scores was inspected; the percentage of individuals with the lowest and highest possible score in each of the factors was recorded, and values greater than 20% were considered as floor and ceiling effects.

Associations between participant characteristics and WEMWBS scores were examined with independent sample t-tests. Although focus groups have been used to evaluate participants’ comprehension and ease of completion (Taggart et al., 2013), previous studies of the WEMBWS have not examined its readability. Therefore we used the Flesch Reading Ease score and Flesch–Kincaid Grade Level function to determine the readability of the questionnaire (Flesch, 1948).

Results

Response rate

The total response rate was 90.7% (1271/1400).

Demographics

1,271 individuals responded to the questionnaire. Respondent characteristics are summarized in Table 1. Participants’ ages ranged between 20 and 68 years, with a mean age (SD) of 31.8 (9.4) years. Mean age was greater for men (34.1 (10.3) years) than women (30.0 (8.1) years). Most of the respondents were doctors (n = 840, 66%) or nurses (n = 218, 17.2%), female (n = 720, 57%), and practicing in Lahore (n = 70%).

Table 1 Demographic characteristics of a sample of 1,271 health care providers in Punjab province, Pakistan, who completed the Warwick–Edinburgh Mental Well-being Scale (WEMWBS).

Variable	Frequency (n)	WEMWBS mean (SD)	
Gender	Male	551 (43.4%)	49.16 (9.16)	
	Female	720 (56.6%)	47.30 (9.59)	
Healthcare profession	Doctor	840 (66.1%)	47.74 (9.57)	
	Nurse	218 (17.2%)	47.74 (9.45)	
	Pharmacist	53 (4.2%)	50.28 (8.35)	
	Physiotherapist	25 (2%)	49.44 (6.90)	
	Dentist	135 (10.6%)	49.93 (9.26)	
City	Lahore	893 (70.3%)		
	Multan	86 (6.8%)		
	Sheikhupura	54 (4.2%)		
	Gujrat	58 (4.6%)		
	Rawalpindi	101 (7.9%)		
	Faisalabad	54 (4.2%)		
	Islamabad	25 (2%)		
Age	≥30	542 (42.6%)	48.87 (9.11)	
	>30	729 (57.4%)	47.54 (9.65)	

There were some differences in respondent age and gender distribution in relation to profession: nurses were almost all (97%) female, whereas among doctors and dentists there was a nearly equal gender representation. 42 of the pharmacists were male (79%), as were 16 (64%) of the physiotherapists. There were no significant age differences between the professional groups except for dentists in the sample, whose mean age was younger than for the other professions (28.5 (6.6) years; P < 0.001).

Descriptive statistics and floor or ceiling effects

Mean (SD) score on the WEMBWS was 48.1 (9.4), and median score was 48. Analysis of item response frequencies showed little evidence of highly skewed distributions, with a mild skewness of −0.31 (0.07) and a kurtosis of 0.42 (0.14). Visual inspection of the histograms and Q-Q plots did not reveal any significant deviation of response distribution from normality. All response categories were used by at least one person for all the scale items. Mean values for individual items ranged from 2.88 (1.1) for “I’ve had energy to spare” to 3.74 (0.99) for “I’ve been feeling confident” (Table 2). A histogram of the distribution of mean scores is shown in Fig. 1. Based on these results, we conclude that the WEMWBS is valid for the Pakistani population is free from any floor or ceiling effects.

Figure 1 Frequency distribution of total scores of Pakistani HCPs on WEMWBS (n = 1,271).

Table 2 Item-level statistics for responses on the Warwick–Edinburgh Mental Well-being Scale in a sample of Pakistani health care providers (n = 1,271).

Statement	Mean	Std. deviation	
I’ve been feeling optimistic about the future	3.36	1.109	
I’ve been feeling useful	3.74	.991	
I’ve been feeling relaxed	3.03	1.085	
I’ve been feeling interested in other people	2.97	1.140	
I’ve had energy to spare	2.88	1.113	
I’ve been dealing with problems well	3.53	.959	
I’ve been thinking clearly	3.59	1.034	
I’ve been feeling good about myself	3.58	1.052	
I’ve been feeling close to other people	3.33	1.053	
I’ve been feeling confident	3.74	.999	
I’ve been able to make up my own mind about things	3.72	.992	
I’ve been feeling loved	3.64	1.038	
I’ve been interested in new things	3.58	1.111	
I’ve been feeling cheerful	3.42	1.022	

As in the UK validation study (Tennant et al., 2007) mean WEMWBS scores were significantly higher for men (49.16) than for women (47.30) (t = 3.486, df = 1,269, P = 0.001) (Table 1).

There were differences in well-being scores between professional groups. Doctors’ and nurses’ mean WEMWBS scores were lower than those for the other professional groups. The difference between WEMWBS score for doctors and nurses combined, compared to the other professional groups combined, was statistically significant (P = 0.002, mean difference 2.22, t = − 3.14, df = 1,269). A difference in well-being score was evident across age groups, with participants aged 30 years and older scoring higher on the WEMWBS than younger participants (P = 0.013, mean difference 1.33, t = 2.48, df = 1,269).

Items 2, 10, 11 and 12 received the most positive ratings by both men and women, whereas items 3, 4 and 5 received the least positive ratings by participants of both genders.

Factor analysis with principal components analysis

The WEMWBS responses were subjected to PCA, and the correlation matrix showed that all variables had at least one correlation coefficient greater than 0.3. The overall KMO measure was 0.83, with individual KMO measures all greater than 0.7, thus indicating a “middling” to “meritorious” classification. Bartlett’s test of sphericity was statistically significant (P < 0.0005), indicating that the data were suitable for factor analysis. Our PCA identified two components with eigenvalues greater than 1, and which explained 42.1% and 8.5% of the total variance. Eigenvalues, visual inspection of Cattell’s scree plot, interpretability criteria and Cronbach’s alpha value indicated that one component should be retained. A single-component solution explained 42.1% of the total variance. Total score was obtained by summing all items. Factor loadings of the rotated component solution for all 14 items of the WEMWBS based on eigenvalues greater than 1 are shown in Table 3. The scree plot is illustrated in Fig. 2.

Figure 2 Scree plot for 14-item WEMWBS.

Table 3 Factor loadings for the 14 items in the Warwick–Edinburgh Mental Well-being Scale in a sample of Pakistani health care providers (n = 1,271).

Statements	Factor loading	
	I	II	
I’ve been feeling confident (Item 10)	.789		
I’ve been feeling good about myself (Item 8)	.776		
I’ve been thinking clearly (Item 7)	.744		
I’ve been able to make up my own mind about things (Item 11)	.722		
I’ve been dealing with problems well (Item 6)	.703		
I’ve been feeling cheerful (Item 14)	.686		
I’ve been feeling close to other people (Item 9)	.670		
I’ve been feeling useful (Item 2)	.637		
I’ve been feeling loved (Item 12)	.625		
I’ve been interested in new things (Item 13)	.578		
I’ve been feeling optimistic about the future (Item 1)	.541		
I’ve been feeling interested in other people (Item 4)	.390	.663	
I’ve had energy to spare (Item 5)	.378	.646	
I’ve been feeling relaxed (Item 3)	.508	.528	
Notes.

Extraction method: principal component analysis.

Rotation method: quartimax with kaiser normalization.

Rotation converged in 3 iterations.

For interested readers, the results of exploratory factor analysis with the principal axis factoring method are provided as Table S1.

Internal consistency and content validity

The WEMWBS consists of 14 items. Cronbach’s alpha for single-factor structure of the WEMWBS was 0.89. Corrected item correlations for all items were greater than 0.3, thus denoting the same construct for all items. Item-total statistics for all items are summarized in Table 4. To further analyze internal consistency, we calculated item-total score correlations for each item, with adjustment for overlap. These correlations ranged between 0.46 and 0.68, which are substantial but not excessive values. The item-total statistics for all items in the WEMWBS are shown in Table 4.

Table 4 Item-total statistics for the 14 items in the Warwick–Edinburgh Mental Well-being Scale in a sample of Pakistani health care providers (n = 1,271).

	Scale mean if item deleted	Scale variance if item deleted	Corrected item-total correlation	Squared multiple correlation	Cronbach’s alpha if item deleted	
I’ve been feeling optimistic about the future	44.75	78.422	.488	.293	.887	
I’ve been feeling useful	44.37	78.627	.548	.350	.884	
I’ve been feeling relaxed	45.08	77.650	.544	.355	.884	
I’ve been feeling interested in other people	45.13	78.590	.462	.292	.888	
I’ve had energy to spare	45.23	79.177	.445	.289	.889	
I’ve been dealing with problems well	44.58	77.894	.616	.447	.881	
I’ve been thinking clearly	44.52	76.722	.631	.490	.880	
I’ve been feeling good about myself	44.53	75.597	.684	.525	.878	
I’ve been feeling close to other people	44.78	76.788	.614	.424	.881	
I’ve been feeling confident	44.37	76.938	.644	.518	.880	
I’ve been able to make up my own mind about things	44.39	77.959	.587	.421	.882	
I’ve been feeling loved	44.47	78.118	.547	.348	.884	
I’ve been interested in new things	44.52	77.251	.550	.352	.884	
I’ve been feeling cheerful	44.69	76.463	.656	.492	.879	

Test-retest reliability

The test-retest reliability evaluated as the intraclass correlation coefficient was 0.921, with a 95% confidence interval of 0.861–0.955, thus indicating a very high degree of reliability.

Readability

The values for Flesch reading ease and FK grade level score were calculated in Microsoft Word 2013. Flesch reading ease score for the WEMWBS was 73.5, and FK grade level score was 4.4, which indicate that the WEMWBS was easy for our participants to read and understand.

Discussion

The present study confirms that in our sample of Pakistani HCPs, the WEMWBS showed reliability in measuring variations in a single-factor, high internal consistency, good validity and high readability. Our findings also indicate no floor or ceiling effects in the response distribution in our study population. Our sample was relatively large and consisted of a diverse range of health care professionals (physicians, surgeons, general practitioners, nurses, pharmacists, dentists and physiotherapists), which is an important strength of this study.

The mean (SD) well-being score in this study was 48.1 (9.4), which is lower than values obtained in general population surveys in the UK. The Health Survey for England identified a mean WEMWBS score for adults of 52.3 (Bridges & Health Survey for England, 2012), and the mean score in a recent population survey in Scotland was 50.0 (Wilson et al., 2015). The median score in our study sample was 48, compared to 51 in the initial validation study (Tennant et al., 2007) and 50 in a large-scale population sample in Northern Ireland (Lloyd & Devine, 2012).

We identified gender differences in well-being scores, as well as some differences between professions and age groups. In this study, male HCPs scored higher on the WEMWBS than females, which is consistent with the findings of UK population studies (Bridges & Health Survey for England, 2012; Wilson et al., 2015).

Items 3, 4 and 5 were the least strongly endorsed by HCPs in our study. According to Taggart et al. (2013), item 4 (“I’ve been feeling interested in other people”) was misinterpreted by Chinese and Pakistani groups in the UK, for whom it implied a sexual context. This may be a potential reason for the less positive response to this item in our study.

Like other studies conducted in the general population of the UK, our analysis confirmed the unidimensional construct of the WEMWBS and high internal consistency (Tennant et al., 2007; Clarke et al., 2011). However, the factor structure was not as clear-cut as in the original validation study (Tennant et al., 2007) and other analyses, with some suggestion of two factors. However, the extent of cross-loading and the modest additional variance explained by two factors/components, together with the lack of explanatory or theoretical underpinning for this structure, indicate that a unidimensional model can be assumed.

A study of Pakistani people in the UK also showed high internal consistency, but as in the current study, the factor structure was less clear given that three factors were evident. Nevertheless, the limited variance explained by the second and third factors support a single-factor model (Taggart et al., 2013).

There was some correspondence between the item-total correlations identified in our study and in the Taggart et al. (2013) study of UK English-speaking Pakistanis. The three items with the lowest item-total correlations in our study (item 1, “I’ve been feeling optimistic about the future”; item 4, “I’ve been feeling interested in other people”; and item 5, “I’ve had energy to spare”) also had the lowest correlations in the UK-Pakistani study, though the markedly low value for item 1 found in the UK sample (0.28) was not evident in our sample of Pakistani HCPs (0.488). In focus groups conducted as part of the UK study, it was found that a quarter of Pakistanis who completed the questionnaire were native Pashtun speakers who were born abroad, and there is no translation for “optimistic” in the Pashto language (Taggart et al., 2013). However, our sample consisted of Punjabi- and Urdu-speaking health care professionals.

The histogram was slightly negatively skewed and kurtotic; however, no floor or ceiling effects were observed in the WEMWBS, in consonance with other validation studies. The WEMWBS may also prove useful for evaluating mental well-being in undergraduates thanks to its easy readability and lucid content, as demonstrated by its Flesch reading ease score and FK grade level score of 4.4. This fact was also supported by favorable comments from the participants in our pilot survey. Based on these findings, we speculate that this scale can be easily used for teenagers, college and university students.

Limitations

This cross-sectional study used a convenience sample of HCPs and so, unlike studies that involve a systematically derived probability sample, the extent to which the respondents are representative of the target population is uncertain. This is likely to be of less importance for psychometric evaluation than the descriptive results, and is partially compensated for by the relatively large sample size and high response rate: 91% of those asked to complete the questionnaire did so, which reduced participation and selection bias.

Comparator instruments were not used alongside the WEMWBS, so further efforts to evaluate criterion validity are advisable.

Translation and back-translation to develop an Urdu language version (the national language of Pakistan) were not done as part of this study, because our sample consisted of HCPs who were all capable of reading and writing in English. So although the WEMWBS has been shown suitable for assessing mental well-being in a diverse group of HCPs, it cannot yet be used for studies within the general population of Pakistan, where the extent of fluency in English is considerably less.

The WEMWBS assesses hedonic and eudaimonic constructs of mental well-being but does not contain any items intended to evaluate in individual’s spiritual and religious well-being—aspects which predominate in Eastern traditions and in Pakistani culture, where religious beliefs and attitudes have emerged as an important coping mechanism (Waqas et al., 2014; Voll, 1992). This potential limitation was also brought to light in a study of ethnic minority groups of Malay, Chinese and Indian Muslim and Hindu origin (Vaingankar et al., 2011). We therefore suggest that the new generation of psychometric instruments should include items designed to evaluate these constructs.

Recommendations

The WEMWBS is particularly useful for monitoring mental well-being levels in HCPs and medical students who work in highly demanding roles and environments. Further studies are required to develop and validate an Urdu version of this scale in the general community. Along with general studies of the range of factors that are associated with mental health and well-being, the WEMWBS may also be useful to evaluate the efficacy of stress reduction programs such as mindfulness programs, relaxation exercises, cognitive therapies, and social support and group therapies which have proved useful in academic and hospital-based settings (Shapiro, Shapiro & Schwartz, 2000).

Conclusion

As far as we are aware, this is the first study designed to validate a scale used to evaluate mental well-being in a Pakistani sample, and is thus an important addition to the limited arsenal of validated psychometric instruments available for this population. The WEMWBS showed excellent psychometric properties in Pakistani HCPs, and appears to be a valid and reliable tool for use among English-speaking Pakistani people.

Supplemental Information

Table S1 Factor loadings for the 14 items in the Warwick–Edinburgh Mental Well-being Scale in a sample of Pakistani health care providers (n = 1271)

Click here for additional data file.

Supplemental Information 1 Datafile in csv format

Click here for additional data file.

The authors thank Professor Dr. Atif Rahman, Professor of Child Psychiatry at the University of Liverpool, United Kingdom, for his valuable suggestions for improving this manuscript. The authors also thank K Shashok (AuthorAID in the Eastern Mediterranean) for improving the use of English in the manuscript.

Additional Information and Declarations

Competing Interests

Author Contributions

Ethics

The authors declare there are no competing interests.

Ahmed Waqas conceived and designed the experiments, performed the experiments, analyzed the data, contributed reagents/materials/analysis tools, wrote the paper, prepared figures and/or tables, reviewed drafts of the paper.

Waqas Ahmad conceived and designed the experiments, performed the experiments, contributed reagents/materials/analysis tools, wrote the paper, reviewed drafts of the paper.

Mark Haddad analyzed the data, contributed reagents/materials/analysis tools, wrote the paper, prepared figures and/or tables, reviewed drafts of the paper.

Frances M. Taggart conceived and designed the experiments, performed the experiments, contributed reagents/materials/analysis tools, wrote the paper, prepared figures and/or tables, reviewed drafts of the paper.

Zerwah Muhammad and Shahzad Ahmed Sami performed the experiments, analyzed the data, contributed reagents/materials/analysis tools, wrote the paper, prepared figures and/or tables, reviewed drafts of the paper.

Muhammad Hamza Bukhari and Sumbul Ejaz performed the experiments, contributed reagents/materials/analysis tools, wrote the paper, reviewed drafts of the paper.

Sayyeda Mehak Batool and Ayesha Hanif performed the experiments, contributed reagents/materials/analysis tools, reviewed drafts of the paper.

Fiza Najeeb performed the experiments, analyzed the data, contributed reagents/materials/analysis tools, prepared figures and/or tables, reviewed drafts of the paper.

Zehra Ali Rizvi performed the experiments, contributed reagents/materials/analysis tools, prepared figures and/or tables, reviewed drafts of the paper.

The following information was supplied relating to ethical approvals (i.e., approving body and any reference numbers):

Ethical approval was obtained from the Ethical Review committee of CMH Lahore Medical College, Lahore Cantt, Pakistan.

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
