# Peer review of "Measuring the well-being of health care professionals in the Punjab: a psychometric evaluation of the Warwick–Edinburgh Mental Well-being Scale in a Pakistani population"

_PeerJ, doi:10.7717/peerj.1264_

## Round 0.1 · original submission · Major Revisions

I believe this manuscript has the potential to be published, but the reviewers have raised a number of comments that should be addressed. Any resubmission should be accompanied by a detailed list of the comments as you interpret them and how you have responded. Should you choose not to revise the manuscript in response to a particular comment, please indicate your rationale.

In addition to the comments made by the reviewers, I have an additional one: exactly how was the convergent validity of the measure established? I would've expected that correlations would've been provided with other, establish measures.

Reviewer 1 ·

Basic reporting

References in text do not conform to template and should be changed.

More detail is required in the Introduction on the WEMWBS.

Proof-read all text for use of English and typos.

Experimental design

The Methods section needs more detail, particularly in relation to the Procedure to ensure it could be replicated.

An explanation as to why PCA rather than FA was chosen.

Validity of the findings

It needs to be made clear throughout that this is NOT a general population study and is, indeed, not generalisable to the wider health professional population given the use of a convenience sample.

Additional comments

Abstract
Not Pakistani population but healthcare professionals. As this was a convenience sample (Line 112) it is necessary to take care when making generalisations to the ‘population’. This is something that needs to be addressed throughout this paper before publication in the journal.
Line 41 – example of these countries (perhaps in footnote?) and references for this?
Line 44 – ‘our society’ – perhaps keep more objective.
Line 59 – you should ensure you differentiate between when you are referring to the general population of Pakistan and the health professional population in Pakistan
Line 62 and 66
You refer to developed countries and Western countries – are these being used interchangeably?
Line 72-73
Although these are not ‘UK’ wide – no inclusion of Northern Ireland or Wales – see Lloyd, K. and Devine, P. Psychometric properties of the Warwick–Edinburgh
Mental Well-being Scale (WEMWBS) in Northern Ireland (2012), Journal of Mental Health, 21(3), 257–263.
Line 71
An explanation as to what aspects of the WEMWBS make it particularly useful for this population apart from the fact that it has been validated in the UK would be useful.
Line 72
As above, this study does not include a sample of the general population but relates only to HCPs.
Line 90
Should be ‘with’ rather than ‘on’ – we have moved away from considering participants as ‘subjects’ that research is carried out ‘on’.
Line 94
Therefore the WEMWBS was presented in English?
What procedure was used? Was this a self-completed questionnaire or, as implied by Line 114, interviewer administered?
Line 123
Perhaps explain the rationale for using PCA rather than FA
Line 154
As before, be specific that this is among a sample of health professionals not the general population.
Line 161
It is also useful to use Monte-Carlo Parallel Analysis to confirm factor structure when using PCA

Table 1
Check sign for second category of age
General
Proof-read thoroughly for use of English and typos.

·

Basic reporting

This is a well written paper describing a well conducted validation of a scale of mental wellbeing in a cross cultural setting.
The figures and tables are well presented and clear
See below in general comments for a few minor comments which could improve readability

Experimental design

The study is original and the methods are sound and well described
There has been ethical approval

It adds value to what is already known and will enable the conduct of further research into the mental wellbeing of health care professionals a topic of current interest and importance.

Validity of the findings

The results are appropriately interpreted with one exception (see below)
The study adds value to what is already known and will enable the conduct of further research into the mental wellbeing of health care professionals a topic of current interest and importance.

Additional comments

This is a well written paper describing a well conducted validation of a scale of mental wellbeing in a cross cultural setting. It adds value to what is already known and will enable the conduct of further research into the mental wellbeing of health care professionals a topic of current interest and importance.
Abstract
Line 24 needs to include the fact that the English rather than Urdu version of the scale that was validated
Materials and methods
Line 114. It is not clear why the student researchers need to undertake a two day workshop in interviewing skills – what interviewing was done in the project?
Line 140. Readability – WEMWBS readability has been assessed before, but as I am struggling to find out in which publication that research is reported it is good that it has been repeated. It would be helpful if the FK grade level score could be translated into a rough reading age for those not familiar with interpreting Flesch Scores Line
Line 157. WEWMBS appears as ’MWEBS’
Line 167.’ Total score was obtained by summing all the items’. This sentence should appear under methods.
Limitations
Line 203 It would be better to say that this study has not assessed validity in the general population (because it was conducted in HCP and in the English version) instead of saying ‘it cannot be used for the general population’. The latter implies that the scale has been shown not to work in the general population which is not true.
Line207 Discussion about the absence of spiritual dimensions of wellbeing in WEMWBS and their relevance in Pakistani culture is valid and appropriate. However there is a question as to whether spiritual wellbeing should be assessed in the same or a different instrument. The concepts are closely related but it might be that they are better addressed in a different instrument. What this study shows is that this mental wellbeing scale works well among HCPs in Pakistan. If spiritual items were necessary to a mental wellbeing scale in this setting it would not have worked that well
Recommendations
Line 215 – see above. The first recommendation does not follow from the research presented (see above) and should be omitted.

---

## Round 0.2 · Minor Revisions

I would be prepared to accept this manuscript, contingent on a minor revision responsive to the reviewer's comments.

Reviewer 1 ·

Basic reporting

The article needs much more proof-reading for typos, missing words, grammar etc. Specific comments are outlined in 'comments to the author'.

Experimental design

No comments

Validity of the findings

No comments

Additional comments

The majority of the issues raised have been dealt with; two things need to be addressed:
In response to the query regarding convergent validity, the authors states ‘Convergent validity has not been mentioned in the revised manuscript and it has also been addressed in the limitation section.’ This is still mentioned in the revised manuscript I have access to, along with reference to criterion validity in the Limitations section. It is still not clear how validity was established and this needs to be addressed.

More proof-reading is required for example spelling: ‘Keyowords’, ‘Chronbach’s’, ‘Lloyed’ in References should be Lloyd, etc.
This needs to be addressed before publication.

---

## Round 0.3 · accepted · Accept

Thanks for your responsiveness.